# Improving Nonlinear Projection Heads using Pretrained Autoencoder Embeddings

## Abstract

This empirical study aims at improving the effectiveness of the standard 2-layer MLP projection head $g(\cdot)$ featured in the SimCLR framework through the use of pretrained autoencoder embeddings. Given a contrastive learning task with a largely unlabeled image classification dataset, we first train a shallow autoencoder architecture and extract its compressed representations contained in the encoder's embedding layer. After freezing the weights within this pretrained layer, we use it as a drop-in replacement for the input layer of SimCLR's default projector. Additionally, we also apply further architectural changes to the projector by decreasing its width and changing its activation function. The different projection heads are then used to contrastively train and evaluate a feature extractor $f(\cdot)$ following the SimCLR protocol, while also examining the performance impact of $Z$-score normalized datasets. Our experiments indicate that using a pretrained autoencoder embedding in the projector can not only increase classification accuracy by up to 2.9 % or 1.7 % on average but can also significantly decrease the dimensionality of the projection space. Our results also suggest, that using the sigmoid and $\tanh$ activation functions within the projector can outperform ReLU in terms of peak and average classification accuracy. When applying our presented projectors, then not applying $Z$-score normalization to datasets often increases peak performance. In contrast, the default projection head can benefit more from normalization. All experiments involving our pretrained projectors are conducted with frozen embeddings, since our test results indicate an advantage compared to using their non-frozen counterparts.

## 1 Introduction

Contrastive learning is a self supervised machine learning paradigm that has proven itself to be useful for Big Data applications, which can capture nearly unlimited amounts of unlabeled data at high speeds and low costs. In such situations, classical supervised learning approaches become infeasible due to missing labels and because they can neither be generated in time nor in sufficient quality. For this reason, despite the groundbreaking success of supervision in the field of machine learning, the research community will continue to investigate new and existing methods for learning representation with a limited amount of labeled data. This paper combines the well understood fundamentals of unsupervised learning using autoencoders with more recent approaches of contrastive learning as offered by the SimCLR framework (Chen et al., 2020). Both autoencoders and SimCLR are designed to be trained without any labeled data, although SimCLR's evaluation protocol requires some amount of labeled samples. This is due to the fact that SimCLR is usually used for training image classification models, whereas autoencoders are often utilized for image generation and reconstruction tasks.

In order to understand how the concepts behind SimCLR and plain autoencoders are combined in this paper, one first has to understand the abstract structure of SimCLR. At its most basic level, SimCLR is a composition of two artificial neural networks, the backbone $f(\cdot)$ followed by the projection head $g(\cdot)$. Instead of focusing on the backbone, which is a deep convolutional neural network responsible for feature extraction, we direct our attention to the projection head, a shallow multilayer perceptron (MLP) consisting of two ReLU activated linear layers. The main role of this architectural building block is to project the features generated by the backbone into a lower dimensional projection space. This latent space and its embeddings can significantly influence the backbone's

ability to learn meaningful representations as the contrastive loss is applied to the outputs of the projector. This highlights the importance of the projection head within the SimCLR framework since backpropagation will always adapt the backbone's weights based on the loss values generated by the compressed projections. Also, compressing certain inputs into more efficient representations can generally be seen as the central task of autoencoders. In fact, we find regular autoencoders to be a special type of MLP that can be trained to reconstruct inputs by optimizing a hidden embedding layer using some reconstruction loss that captures the difference between the encoder's input and the decoder's output. Based on these observations, we expect to increase the quality of contrastively learned representations by exchanging the randomly initialized input layer of SimCLR's default projector with a pretrained autoencoder embedding.

From a practical point of view, it is well known that deeper and wider backbones usually improve representation quality and downstream classification accuracy. In contrast, it is generally not well understood, why the backbone benefits from using a nonlinear projection head. Chen et al. (2020) introduce the projector after empirically observing significant increases in model accuracy by more than 10 %. From this the authors conclude that the hidden layer before the projection head learns better representations than the layer after. The authors also conjecture that including a projection head increases the backbone's ability to aggregate and retain information that is valuable for downstream tasks. However, the rational given for this phenomenon only explains why the backbone's learned representations are of higher quality but not how the projector is able to generate them in the first place. This lack of reasoning becomes obvious when comparing the proposed end-to-end optimization of both the backbone and the projector to only training the backbone without any projection head. Despite some open questions about the projector's precise mode of operation, its contribution to the success of SimCLR and related methods remains undeniable. We therefore conduct experiments as part of this paper to investigate the effectiveness of pretrained autoencoder embeddings within SimCLR's default projector. The main contributions of our paper are summarized as follows:

1. We demonstrate on five well-known image classification datasets that exchanging the input layer of SimCLR's default projector with a pretrained autoencoder embedding reliably increases linear classification accuracy.

2. We show that activating the projector with the sigmoid or $\tanh$ functions can outperform ReLU in terms of peak and average accuracy.

3. We observe that pretrained autoencoder embeddings can drastically reduce the width of the projector as well as the dimensionality of the projection space while increasing average accuracy.

4. We find that freezing the weights within the pretrained embedding layers almost always increases classification accuracy.

5. We note that $Z$-score normalization of datasets drastically impairs the training of autoencoder embeddings.

6. To ensure reproducibility of our results, we publish the training and evaluation code together with CSV listings of all training runs on GitHub: `https://anonymous.4open.science/r/simclr-ae`

The rest of the paper is organized as follows: Section 2 discusses related work. Next, we start to formally introduce our methods in Section 3, which contains an overview about the SimCLR framework, basic autoencoders, and our proposed architecture.In Section 4 we present and compare our results to SimCLR's default projector and discuss them in Section 5. Finally, Section 6 concludes our paper by summarizing our work and giving interesting ideas for future research.

## 2 RELATED WORK

Literature review does not reveal any references to publications dealing explicitly with pretraining the nonlinear projection head used in the SimCLR framework. A few papers aim at explaining the role and the effects of the projector in more detail. These include the study conducted by Jing et al. (2022) who are trying to understand dimensional collapse in contrastive self-supervised learning (SSL). The authors first show that dimensional collapse can also happen in contrastive learning despite the use of positive and negative samples. For SimCLR, dimensional collapse of the representation space usually occurs when the backbone $f(\cdot)$ is trained without a projection head ($g = \mathrm{id}$).

Instead of improving the projection head itself, the authors propose an alternative to SimCLR called DirectCLR, a contrastive learning method that does not rely on a trainable projection head but uses a fixed low rank diagonal projector instead. DirectCLR does not try to prevent dimensional collapse by using a trainable projector but rather through direct optimization of the backbone. The proposed approach outperforms SimCLR when using a 1-layer linear projector by 1.6 % but fails to match the performance of SimCLR's default 2-layer nonlinear projection head by 3.8 %.

In contrast to Jing et al. (2022), the work of Gupta et al. (2022) aims at understanding and improving the role of the projection head in self-supervised learning. The authors conjecture that the projection head implicitly learns to choose a subspace of features on which the contrastive loss is applied. This subspace selection is believed to address the disadvantages of the contrastive loss function which are mainly introduced by sub-optimal data augmentations. It is also assumed that implicit subspace selection can enable the projection head to generate embeddings which minimize contrastive loss, while allowing the backbone to learn meaningful representations. Based on these observations, the authors argue that data-dependent subspace selection should be part of the loss function used in self-supervised learning. This is why the authors reformulate SSL as a bilevel optimization problem where at each training step, the first optimization problem is to select the best subspace for the contrastive loss by optimizing the projection head while the second optimization performs gradient descent on the backbone. As a result, a contrastive loss function is proposed which treats the projector as part of the criterion and not only as an architectural component. The authors continue to empirically demonstrate that a trainable projector generally increases performance and can therefore be viewed as an improved optimization scheme for SSL. This result contradicts the general notion of DirectCLR (Jing et al., 2022), where backbone features are optimized directly using a non-trainable projector.

Since contrastive learning is not only limited to SimCLR and its nonlinear projection head, Gui et al. (2023) address the question why linear projectors affect the generalization properties of learned representations. The authors find the heuristics in the existing literature to be insufficient for answering this question, especially in the case of linear projection heads. Since discarding the projection head after training is a common approach in contrastive learning, the authors assume projectors to act as a buffer component that protects the representation space from distortions induced by loss minimization carried out on the outputs of the projector. For nonlinear projection heads, this assumption shows significant overlap with the conjecture made by Chen et al. (2020) w. r. t. the superiority of the layer before the projector. This may be rooted in information loss triggered by contrastive loss: i. e. , statistical shrinkage and expansion, i. e. a reduction or increase in the effects of sampling variation induced by the contrastive loss on the projector, are highly correlated with downstream accuracy. Expansion decreases test error, while shrinkage reduces signals and increase test error.

Xue et al. (2024) theoretically investigate the benefits of a projection head for representation learning. The authors also observe that linear models progressively assign weights to features as they propagate through the layers of the backbone. Based on this observation, the authors state that features in deeper layers of the backbone are represented more unequally. Furthermore, these representations not only turn out to be more specialized towards the pretraining objective but are believed to be additionally distorted through the use of nonlinear activations within the layers of the backbone. The authors argue that this allows deeper layers to learn features which are entirely absent in the outputs of the projector. As a major result, it is stated that "projection heads provably improve the robustness and generalizability of representations, when data augmentation harms useful features of the pretraining data, or when features relevant to the downstream task are too weak or too strong in the pretraining data". The authors show that these findings also apply to supervised contrastive learning and regular supervised learning where lower layers can also learn subclass-level features that are not represented in the final layer. Finally, it is noted that these findings "demonstrate how representations before the final representation layer can significantly reduce class/neural collapse". In current literature, this effect is well observed for both nonlinear and linear projection heads (Chen et al., 2020), although DirectCLR (Jing et al., 2022) suggests linear projectors to be outdated.

In summary, much of the current literature focuses on understanding the usefulness of projection heads for contrastive learning applications mainly from a theoretical point of view. We conclude from our literature review, that projection heads do increase the quality of contrastively learned representations in the backbone. Here, linear projection heads are significantly inferior to nonlinear projectors and that the advantage gained by linear projectors can also be experienced by directly optimizing the backbone. In contrast to linear projectors, there is no compelling evidence to sug-

gest that nonlinear projection heads are ineffective in contrastive learning applications. In fact, research indicates that nonlinear projectors continue to provide an easy and effective building block for achieving peak downstream performance while reducing the impact of model collapse.

## 3 METHOD

### 3.1 THE SIMCLR FRAMEWORK

SimCLR is a contrastive learning framework by Chen et al. (2020). It allows for learning representations from input images without the use of labeled data. The SimCLR framework builds upon extensively researched and carefully parameterized data augmentation pipelines used to create a positive contrastive image pair $(\tilde{\boldsymbol{x}}_i = t(\boldsymbol{x}), \tilde{\boldsymbol{x}}_j = t'(\boldsymbol{x}))$ for each dataset sample $\boldsymbol{x}$. Here $t$ and $t'$ are two transformations which are randomly sampled from the space $\mathcal{T}$ of particularly effective transformations. These transformations include random cropping and flipping as well as the random application of color jitter, grayscaling and dynamic Gaussian blurring where the kernel size adapts to changes in image resolution.

The composite and easy to implement architecture of SimCLR features an interchangeable feature extractor $f$ (backbone), followed by a ReLU activated 2-layer nonlinear projection head $g$ (projector). The backbone can be any neural network, although in practice, convolutional architectures like ResNet are widely used, especially for image classification tasks. Given an augmented pair of images $(\tilde{\boldsymbol{x}}_i, \tilde{\boldsymbol{x}}_j)$, SimCLR first generates two separate intermediate representations $\boldsymbol{h}_i, \boldsymbol{h}_j$ by applying the feature extractor to both $\tilde{\boldsymbol{x}}_i$ and $\tilde{\boldsymbol{x}}_j$. These representations are then mapped via the projector into a lower dimensional projection space:

$$\boldsymbol{z}_i = g(\boldsymbol{h}_i), \ \boldsymbol{z}_j = g(\boldsymbol{h}_j) \tag{1}$$

The SimCLR framework is trained using the Normalized Temperature-scaled Cross Entropy Loss (NT-Xent),

$$\ell_{i,j} = -\log \frac{\exp(\mathrm{sim}(\boldsymbol{z}_i, \boldsymbol{z}_j)/\tau)}{\sum_{k=1}^{2N} \mathbf{1}_{[\mathrm{k}\neq\mathrm{i}]} \exp(\mathrm{sim}(\boldsymbol{z}_i, \boldsymbol{z}_k)/\tau)} \tag{2}$$

a variant of the InfoNCE loss (van den Oord et al., 2018), which does not require explicit sampling of negative examples for a given positive pair. Instead, given a batch of $N$ samples, all $2N - 2$ augmented samples (excluding the positive pair) are treated as negative examples. Research by Chen et al. (2020) suggests, that this approach requires large batch sizes of up to 4096 to reach maximum efficacy. The NT-Xent loss shown in Equation 2 is calculated by taking the negative natural logarithm of the normalized exponential function (softmax), which is applied to the cosine similarities $\mathrm{sim}(\cdot)$ of the projections $\boldsymbol{z}_i, \boldsymbol{z}_j$ and scaled by a temperature parameter $\tau \in [0, 1]$.

In summary, the main idea behind SimCLR's image augmentations and its use of a contrastive loss function is to force the backbone $f$ to learn fundamental representations that are invariant to a wide range of transformations. This is achieved by minimizing the distance between the projections $\boldsymbol{z}_i, \boldsymbol{z}_j$ of positive pairs while at the same time increasing it to the projections of negative examples from the same batch.

### 3.2 AUTOENCODERS

An autoencoder (Hinton & Salakhutdinov, 2006) is a type of multilayer perceptron consisting of an encoder and a decoder. The encoder can be formalized as a mapping $E$ from an input space $\mathcal{X}$ into the embedding space $\mathcal{Z}$, where both $\mathcal{X} = \mathbb{R}^m$ and $\mathcal{Z} = \mathbb{R}^n$ are often euclidean spaces with $m \geq n$. For an input vector $\boldsymbol{x} \in \mathcal{X}$ the output $E(\boldsymbol{x})$ of the encoder is called its embedding or latent vector $\boldsymbol{z} \in \mathcal{Z}$. The decoder $D$ is usually the mirror image of the encoder. It maps the encoder's outputs $\boldsymbol{z}$ from the embedding space $\mathcal{Z}$ back into the space of decoded messages $\mathcal{X}$. The autoencoder $A$ can therefore be written as a function composition of $E$ followed by $D$, mapping input $\boldsymbol{x} \in \mathcal{X}$ via some $\boldsymbol{z} \in \mathcal{Z}$ to its reconstructed output $\boldsymbol{x}' \in \mathcal{X}$:

$$A : \mathcal{X} \rightarrow \mathcal{X}, \ \boldsymbol{x} \mapsto D(E(\boldsymbol{x})) = \boldsymbol{x}' \tag{3}$$

The objective of an autoencoder is to learn compressed representations from a training dataset that allow the decoder to reconstruct new inputs with as little error as possible. In doing so, the autoencoder learns efficient codings for inputs which are embedded into a lower dimensional latent space. Autoencoders are traditionally trained without labeled data in a purely unsupervised manner.

A basic autoencoder embedding is a single hidden layer whose weights store the compressed representations of the encoder's inputs. This embedding layer is trained by minimizing some reconstruction loss, i. e. a quality function $d\colon \mathcal{X} \times \mathcal{X} \to \mathbb{R}_{\geq 0}$ that measures the reconstruction quality $d(\boldsymbol{x}, \boldsymbol{x}')$ of the encoder's input $\boldsymbol{x} \in \mathcal{X}$ compared to the decoder's output $\boldsymbol{x}' \in \mathcal{X}$. A widely used quality function for optimizing autoencoders is the squared $L^2$ norm, which is also known as the mean squared error (MSE) of the input $\boldsymbol{x}$ and its reconstruction $\boldsymbol{x}'$:

$$d(\boldsymbol{x}, \boldsymbol{x}') = ||\boldsymbol{x} - \boldsymbol{x}'||_2^2 = \frac{1}{m} \sum_{i=1}^{m} (x_i - x_i')^2 \tag{4}$$

Autoencoders can be trained using any mathematical optimization algorithm, although in practice, gradient descent is the most common.

### 3.3 PRETRAINING AUTOENCODER EMBEDDINGS

The pretrained autoencoder embeddings used in this paper are generated by training a basic 4-layer autoencoder on five well-known image classification datasets (see Table 1) using gradient descent and MSE loss. For this part of our research, we use the base implementation of the autoencoder provided by Chadebec et al. (2022) through the `pythae` Python library.

For each of the five datasets, we train three embeddings in a 70/10/20 train, validation and test split using varying latent dimensions of 128, 256 and 512. This equates to a total of 15 different pretrained autoencoder embeddings. Dataset samples are either used in their native resolution or scaled down to a maximum of $128 \times 128\,\text{px}$ in order to reduce both the number of trainable parameters and GPU memory consumption. We do not apply $Z$-score normalization to the datasets as previous experiments with normalization resulted in validation loss becoming unresponsive.

All autoencoder embeddings are trained in parallel on three NVIDIA V100 GPUs with a fixed batch size of 100 and training times ranging from 25 to 50 minutes each, depending on dataset and embedding size. Each training run is configured analogous to Chadebec et al. (Chadebec et al., 2022, p. 25) to last for 100 epochs while using the `Adam` optimizer with an initial learning rate of $10^{-4}$ and no weight decay. We parameterized the `ReduceLROnPlateau` learning rate scheduler to monitor the validation loss with a patience of 10 epochs and a reduction factor of 0.5. At the end of each training run, only the best performing autoencoder checkpoint is saved based on its validation loss.

### 3.4 BUILDING PROJECTORS FROM AUTOENCODER EMBEDDINGS

Using the trained autoencoders, we replace the input layer of SimCLR's default 2-layer MLP projector with the encoder's pretrained embedding layer. This is achieved by matching the number of input features of the pretrained embedding layer to the number of input features of the projector. Both input dimensions depend solely on the number of output features generated by the backbone. The weights within each pretrained embedding layer are frozen due to previous experiments showing an improvement in the backbone's downstream performance.

We continue to modify the standard projector by replacing its ReLU activation with SiLU (Sigmoid Linear Unit), sigmoid and `tanh`. As with SimCLR, our modified projector is activated using the output features generated by its linear input layer. For this reason, the number of input features in the projector's output layer must match the number of output features produced by the input layer. In order to avoid mismatches in layer dimensions during our experiments, we choose to programmatically apply these modifications to the projector's architecture.

Finally, we also change the dimensionality of the projector's embedding space by varying the number of output features in the output layer. Analogous to SimCLR, we limit the maximum number of features present in the final projections $\boldsymbol{z}_i, \boldsymbol{z}_j$ to 128. We additionally examine the influence of projecting the intermediate representations $\boldsymbol{h}_i, \boldsymbol{h}_j$ into lower dimensional latent spaces with 64 and

32 dimensions respectively. To ensure a fair comparison, we also construct identically structured projectors using randomly initialized layers for every custom projector with a pretrained input layer.

### 3.5 END-TO-END TRAINING OF BACKBONE AND PROJECTORS

In addition to the projection heads constructed in the previous section, we choose ResNet34 as our backbone over the much deeper ResNet50 architecture used in the original SimCLR paper. This choice is based on the fact, that the autoencoder implementation of Chadebec et al. (2022) uses an embedding layer with 512 input features, matching the number of output features of ResNet34. ResNet variants deeper than ResNet34 increase their final feature dimensions from 512 to 2048 (He et al., 2016) and would therefore render all custom projection heads incompatible.

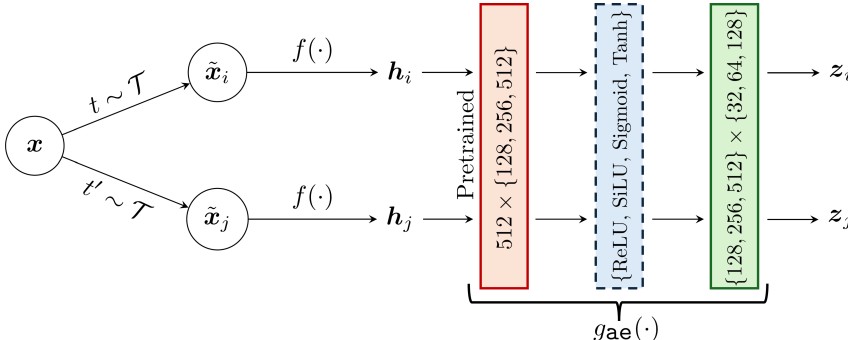

Figure 1: Our modified SimCLR architecture with ResNet34 as $f(\cdot)$, followed by a nonlinear projector $g(\cdot)$ with a pretrained autoencoder embedding (red) as input layer.

We train our modified SimCLR architecture as depicted in Figure 1 using mostly the same set of hyperparameters as proposed by Chen et al. (2020). In order to compensate for the shallower backbone, we choose to train for 150 epochs while using Stochastic Gradient Descent (SGD) as the optimizer with an initial learning rate of 30 % the batch size divided by 256 and momentum set to 0.9. We choose a weight decay of $10^{-5}$ and dynamically adapt the learning rate using the Cosine Annealing scheduler with parameters $T_{\max} = 150$ and $\eta_{\min}$ set to a fiftieth of the initial learning rate. We parameterize the NT-Xent loss using a fixed temperature value of $\tau = 0.5$. Training is conducted using minibatch sizes that allow for at least one full validation batch. For larger datasets like CIFAR10 and CIFAR100, we use a batch size of 1280 (see Table 1). All training runs use the same 70/10/20 dataset splits which were created in Section 3.3 to train the autoencoder embeddings. Individual dataset samples $x$ are augmented into a contrastive image pair $\tilde{x}_i, \tilde{x}_j$ using two randomly sampled transformations $t, t'$ from SimCLR's default transformation space $\mathcal{T}$.

Table 1: Summary of image classification datasets used in our experiments together with their basic characteristics including SimCLR training resolutions and batch sizes.

| Dataset | Imagenette | STL10 | CIFAR10 | FGVCAircraft | CIFAR100 |
|---|---|---|---|---|---|
| Classes | 10 | 10 | 10 | 30 | 100 |
| Type | RGB | RGB | RGB | RGB | RGB |
| Batch size | 392 | 500 | 1280 | 666 | 1280 |
| Resolution | $128 \times 128$ | $96 \times 96$ | $32 \times 32$ | $128 \times 128$ | $32 \times 32$ |

Unlike during the training of the autoencoders, we now allow training runs to normalize datasets prior to augmentation using the standard score or $Z$-score method. We train and evaluate 144 different SimCLR models for each of the five image classification datasets with 72 different projection heads of which 36 include a pretrained autoencoder embedding. As part of these training runs, we also examine the performance impact of the following influencing factors:

- Three latent dimensions (128, 256, 512) for the projector's input layer.
- Four different activations (ReLU, SiLU, sigmoid, tanh) of the projector.
- Three latent dimensions (32, 64, 128) for the final projections.

- $Z$-score normalization of datasets prior to training and evaluation.

## 3.6 Evaluating Learned Representations

For each trained backbone $f$, we evaluate the quality of its contrastively learned representations by training a logistic regression model on the intermediate representations $h$ using a labeled test split. This approach follows the standard evaluation protocol of SimCLR and related methods. The regression model is represented by a single linear layer whose number of input features equals the dimensionality of $h$. The number of its output features is dynamically determined by the the the number of classes present in the test split. We train the logistic regression model for 50 epochs with no weight decay, a learning rate of $10^{-3}$ and a minibatch size of 32. We apply standard softmax to the model's logits to obtain class probabilities.

In order to train and evaluate the logistic regression model, we generate feature datasets for each SimCLR model by first discarding the projection head and then predicting representations $h = f(x)$ for samples $x$ from the test split using only the trained backbone. We then split each feature dataset via a 70/10/20 scheme into new train, validation and test splits. After training and evaluating the regression model on these datasets, we finally report its peak classification accuracy (Acc@1) while assuming linear feature separability to be a valid proxy for the quality of learned representations.

## 4 Results

We begin by summarizing the best test accuracies (Acc@1) achieved by our custom projection heads (`ae`) and the default 2-layer MLP (`mlp`) used in the SimCLR framework on five well-known image classification datasets (see Table 2). A first expected observation is that classification accuracies decrease with increasing number of classes, regardless of the type of projection head used. However, this correlation is observed to be strongly nonlinear, meaning the accuracies achieved on related datasets like CIFAR10 and CIFAR100, where the number of classes increases tenfold, worsen only by about a factor of three. Our absolute best result is recorded on the STL10 dataset where our custom projector outperforms the standard 2-layer MLP by 2.9 %. On average, we achieve an improvement in classification accuracy of 1.74 % across all datasets.

Table 2: Peak test accuracies out of all training runs in relation to varying four influencing factors.

| Dataset | $\dim(g)$ | | **Proj. Activation** | | $\dim(z_{i;j})$ | | **Normalized** | | **Acc@1** | |
|---|---|---|---|---|---|---|---|---|---|---|
| Head | `ae` | `mlp` | `ae` | `mlp` | `ae` | `mlp` | `ae` | `mlp` | `ae` | `mlp` |
| Imagenette | 256 | 256 | Sigmoid | Sigmoid | 128 | 128 | False | False | **89.0 %** | 88.7 % |
| STL10 | 256 | 256 | Sigmoid | Sigmoid | 128 | 128 | False | True | **72.3 %** | 69.4 % |
| CIFAR10 | 128 | 256 | Tanh | Sigmoid | 128 | 64 | False | False | **66.5 %** | 64.9 % |
| FGVCAircraft | 256 | 256 | ReLU | ReLU | 128 | 32 | True | False | **25.7 %** | 24.0 % |
| CIFAR100 | 512 | 512 | SiLU | ReLU | 64 | 64 | False | False | **23.6 %** | 21.5 % |

Table 2 also summarizes the best evaluation results obtained after altering the values of the four major influencing factors listed in Section 3.5. We choose $\dim(g)$ to denote the number of output features in the input layer of the projector $g$ and $\dim(z_{i;j})$ to represent the dimensionality of the final projections, i.e. the number of output features of the projector. Our results listed in Table 2 indicate, that in order to achieve peak classification accuracy across all datasets, the number of output features in the output layer of our custom projector has to be increased in some cases when compared to the default 2-layer MLP. On average, however, we see a significant reduction in projector width when using a pretrained input layer as outlined in Table 4.

We also observe, that activating the projector using either the sigmoid or $\tanh$ function outperforms ReLU on all datasets with ten classes, whereas more complex datasets like FGVCAircraft and CIFAR100 work better with linear units. Counterintuitively, we find $Z$-score normalization to reduce peak classification accuracy for both projectors across a majority of different datasets.

In order to further analyze the performance of the different projection heads, we report key statistical indicators for the distribution of test accuracies across all runs grouped by projector type and dataset. The upper half of Table 3 shows the statistics for the training runs in which $Z$-score normalization was performed, whereas the lower half lists the results without dataset normalization.

Statistical analysis reveals, that by using our custom projector the interquartile range (midspread) of test accuracies is almost always reduced, indicating a general improvement in model stability. This observation can also be made for datasets where the absolute range and standard deviation of test accuracies is larger, hinting at an increased number of outliers when using the `ae` projector. In addition to the maximum values, the mean result also suggest significant improvements in average classification accuracy when using our pretrained projectors over the default 2-layer MLP. We find this observation to be true for all datasets, even the ones with 30 and 100 classes.

Table 3: Statistical indicators for the distribution of test accuracies recorded on different image classification dataset with (top) and without (bottom) $Z$-score normalization.

| Dataset | Imagenette | | STL10 | | CIFAR10 | | FGVCAircraft | | CIFAR100 | |
|---|---|---|---|---|---|---|---|---|---|---|
| Head | ae | mlp | ae | mlp | ae | mlp | ae | mlp | ae | mlp |
| min. | 41.9 % | **68.4 %** | 42.6 % | **44.4 %** | **56.8 %** | 44.6 % | **20.4 %** | 20.1 % | **8.3 %** | 3.2 % |
| max. | 88.5 % | 88.5 % | **70.8 %** | 69.4 % | **66.3 %** | 64.1 % | **25.7 %** | 23.9 % | **23.3 %** | 21.5 % |
| avg. | **83.6 %** | 83.1 % | **64.4 %** | 59.7 % | **62.6 %** | 58.5 % | **23.4 %** | 22.1 % | **19.0 %** | 16.2 % |
| range | 46.6 % | **20.1 %** | 28.2 % | **24.9 %** | **9.4 %** | 19.5 % | 5.3 % | **3.8 %** | **15.0 %** | 18.3 % |
| IQR | **3.2 %** | 3.3 % | **3.8 %** | 6.6 % | **3.3 %** | 3.8 % | 1.7 % | **1.0 %** | **3.1 %** | 4.7 % |
| SD | 8.0 % | **4.3 %** | 6.8 % | **5.6 %** | **2.4 %** | 3.6 % | 1.2 % | **0.9 %** | **3.1 %** | 4.2 % |
| min. | **66.9 %** | 62.5 % | **39.1 %** | 32.3 % | 44.4 % | **48.8 %** | **20.9 %** | 20.6 % | **5.6 %** | 3.2 % |
| max. | **89.0 %** | 88.7 % | **72.3 %** | 67.8 % | **66.5 %** | 64.9 % | **25.2 %** | 24.0 % | **23.6 %** | 21.5 % |
| avg. | **85.0 %** | 82.8 % | **65.1 %** | 58.8 % | **61.3 %** | 58.7 % | 23.0 % | 22.3 % | **19.4 %** | 16.6 % |
| range | 22.2 % | 26.1 % | 33.1 % | 35.5 % | 22.2 % | **16.1 %** | 4.4 % | **3.5 %** | 18.0 % | 18.4 % |
| IQR | **2.2 %** | 3.2 % | **4.8 %** | 10.2 % | **2.9 %** | 5.5 % | 1.4 % | **1.1 %** | **2.6 %** | 4.0 % |
| SD | **3.8 %** | 6.1 % | **6.0 %** | 8.0 % | 4.4 % | **4.0 %** | 1.0 % | **0.8 %** | **3.1 %** | 4.1 % |

In Table 4 we examine the influence of varying the width of the projector by changing the number of output features in the input and output layers. We deliberately constrain our experiments to network width rather than depth to avoid any major architectural changes compared to the default 2-layer MLP. The average test accuracies listed in the upper half of Table 4 are obtained by varying the number of output features in the projector's output layer, whereas the bottom half contains the average results achieved by changing the number of output features in the projector's output layer. We monotonically decrease the output features in the input layer from 512 to 128 and in the output layer from 128 to 32 in order to generate contraction within the projector. This approach is consistent with the standard SimCLR approach, but instead of limiting the final embeddings $z_{i;j}$ to 128 feature dimensions, it allows for even smaller projection spaces. Our experiments indicate, that using a pretrained autoencoder embedding inside the projector can reliably decrease the number of output features needed in both layers while achieving better classification results. On average, `ae` training runs using 128 or 256 as $\dim(g)$ and 32 or 64 for $\dim(z_{i;j})$ achieve higher test accuracies on all datasets compared to their wider `mlp` counterparts.

In Table 5 we summarize the average test accuracies as a function of projector activation. The results show, that on average, our modified `ae` projector prefers the classical `tanh` activation function over modern linear units like ReLU and SiLU. We find this observation to be true for almost all datasets, although for reaching peak performance on Imagenette and STL10, sigmoid is still preferred (see Table 2). It should be noted that on average, the `mlp` projector also works best with the sigmoid activation function when trained on the Imagenette, CIFAR10 and FGVCAircraft datasets.

Table 4: Average test accuracies as a function of projector width. The top half contains accuracies for different $\dim(g)$. The bottom half lists results for varying $\dim(z_{i;j})$.

| Dataset | Imagenette | | STL10 | | CIFAR10 | | FGVCAircraft | | CIFAR100 | |
|---|---|---|---|---|---|---|---|---|---|---|
| Head | ae | mlp | ae | mlp | ae | mlp | ae | mlp | ae | mlp |
| 128 | **84.8 %** | 82.3 % | **67.0 %** | 59.1 % | **60.9 %** | 57.8 % | **23.5 %** | 22.2 % | **19.4 %** | 16.2 % |
| 256 | 83.4 % | 82.1 % | 66.9 % | 58.5 % | 62.2 % | **59.4 %** | 23.2 % | **22.4 %** | 18.4 % | **17.5 %** |
| 512 | 84.8 % | **84.5 %** | 60.4 % | **60.0 %** | 62.8 % | 58.5 % | 22.9 % | 22.1 % | **19.7 %** | 15.5 % |
| 32 | 82.1 % | 81.8 % | 61.8 % | 56.4 % | **60.9 %** | 55.7 % | **22.8 %** | 22.2 % | **18.2 %** | 15.5 % |
| 64 | **85.6 %** | 81.9 % | **66.1 %** | 58.0 % | 62.7 % | **60.3 %** | 23.2 % | **22.5 %** | 19.6 % | 16.1 % |
| 128 | 85.2 % | **85.1 %** | 66.4 % | **63.3 %** | 62.3 % | 59.8 % | **23.5 %** | 21.9 % | **19.8 %** | **17.6 %** |

Table 5: Average test accuracies over all runs as a function of projector activation.

| Dataset | Imagenette | | STL10 | | CIFAR10 | | FGVCAircraft | | CIFAR100 | |
|---|---|---|---|---|---|---|---|---|---|---|
| Head | ae | mlp | ae | mlp | ae | mlp | ae | mlp | ae | mlp |
| ReLU | 85.0 % | 84.0 % | 64.5 % | **59.9 %** | **62.9 %** | 59.3 % | 23.0 % | 22.0 % | 20.1 % | **19.3 %** |
| SiLU | 85.3 % | 84.5 % | 64.7 % | 59.7 % | 61.1 % | 59.1 % | 23.2 % | 22.1 % | 19.6 % | 18.6 % |
| Sigmoid | 81.8 % | **84.7 %** | 64.6 % | 58.5 % | 61.2 % | **59.4 %** | 23.1 % | **22.8 %** | 16.6 % | 13.4 % |
| Tanh | **85.3 %** | 78.4 % | **65.3 %** | 58.9 % | 62.6 % | 56.6 % | **23.4 %** | 22.0 % | **20.4 %** | 14.3 % |

In Table 6 we finally report our initial test results comparing the performance impact of frozen and non-frozen autoencoder embeddings. In the non-frozen setting, the weights within each pretrained embedding layer are allowed to change in the course of SimCLR's end-to-end training. Separate training runs, in which we use sigmoid as the projector's activation function and do not apply $Z$-score normalization to the datasets, clearly show an increase in peak and average test accuracy after freezing the weights within the pretrained autoencoder embedding. We argue that these results justify the decision to carry out all `ae` training runs using only frozen embeddings.

Table 6: Comparison of peak and average test accuracies achieved by our frozen and non-frozen projection heads (`ae`) using sigmoid activation without normalization.

| Dataset | Imagenette | | STL10 | | CIFAR10 | | FGVCAircraft | | CIFAR100 | |
|---|---|---|---|---|---|---|---|---|---|---|
| Acc@1 | max. | avg. | max. | avg. | max. | avg. | max. | avg. | max. | avg. |
| Frozen | **89.0 %** | 85.2 % | **72.3 %** | 64.1 % | **65.4 %** | 59.8 % | **25.2 %** | 23.3 % | **22.6 %** | 17.5 % |
| Not frozen | 87.9 % | 86.2 % | 71.3 % | 60.5 % | 63.7 % | 56.4 % | 24.3 % | 23.1 % | 21.9 % | 17.1 % |

## 5 DISCUSSION

Our experiments demonstrate that the SimCLR framework can benefit from using pretrained autoencoder embeddings over a randomly initialized input layer to its nonlinear projection head. This finding does therefore confirm our initial hypothesis outlined in Section 1. We explain the success of our approach by the fact that pretraining even a simple autoencoder on a given dataset can capture meaningful representations in its embedding layer. These features can then be exploited within the SimCLR framework by offering a priori knowledge about the high-level structure of the training data. By including these predetermined features into the projector, subsequent training iterations can benefit immediately from these representations. This effect cannot be achieved using the standard 2-layer MLP as the weights of its input layer are randomly initialized. Due to the NT-Xent loss being applied only to the projections $z_i, z_j$, it is reasonable to assume that using meaningful high-level representations within the projector can guide gradient descent more effectively, especially in the early stages of training.

Considering the architectural simplicity of our autoencoder, we find the absolute and average improvements in classification accuracy to be satisfactory, especially in the case of datasets containing a large number of classes. It should also be noted, that in our experiments we neither adapt the architecture of the backbone nor the projection head to compensate for such datasets or different sample resolutions. Regardless of dataset complexity, we assume absolute improvements in classification accuracy to be achievable through deeper and wider backbones trained for longer periods of time. We also suspect that pretraining more advanced autoencoder architectures could further increase the projector's overall performance.

Considering the main purpose of autoencoders, we conjecture that the same features which are suitable for image reconstruction tasks, can also enhance contrastive representation learning as demonstrated in this paper. Additional benefits of our approach can be observed in architectural efficiency based on the decreased number of output features required for the pretrained projector to surpass the standard MLP in average classification accuracy (see Table 4). This finding suggests that the implicit training of the standard MLP projector within the SimCLR framework is often less efficient than the independent pretraining of an autoencoder embedding.

The outstanding performance of the sigmoid activation function in the projector may be enabled by its shallow architecture, which does not offer enough hidden layers for the vanishing gradient

problem to arise. Due to the fact that the comparatively deep ResNet34 backbone is only using ReLU as its activation function, the backwards pass is unable to generate enough derivatives in the range $[0, 1)$, which multiplied together could cause the gradient to vanish. By using the sigmoid activation in the projector, the calculated gradient response at the beginning of each backward pass is likely dampened by its derivative being always less than one. This however is unlikely to trigger a vanishing gradient on its own.

## 6 CONCLUSIONS

In this paper we investigate the hypothesis whether pretrained autoencoder embeddings are able to improve the performance of the standard 2-layer MLP projection head used in the SimCLR framework. Since, from a theoretical point of view, the exact role of the classical nonlinear projector remains unclear, we decide to empirically validate our hypothesis on five well-known image classification datasets. After conducting our experiments, we are able to draw the following conclusions:

- **Increase in accuracy**: Using a pretrained and frozen autoencoder embedding in SimCLR's default projector can increase classification accuracy by up to $2.9\,\%$ or $1.7\,\%$ on average (see Table 2).

- **Architectural efficiency**: On average, a pretrained autoencoder embedding reliably decreases projector width and the dimensionality of the associated projection space (see Table 4).

- **Projector activation**: On datasets with ten classes, both projector types achieve their highest classification accuracies when trained with the sigmoid or $\tanh$ activation function (see Tables 2, 5).

- **Projector stability**: By freezing the weights within the pretrained embeddings, we demonstrate the long lasting efficacy of our projector across multiple training epochs. If we allow the weights to be changed during training, a noticeable decrease in test accuracy is observed (see Table 6).

The results summarized above open up the possibility for a wide range of future research activities. A particularly interesting research question would be whether more advanced autoencoder architectures can further improve test accuracy for datasets with a large number of classes. Other research ideas include exploring different ways of pretraining the projector's input layer or even the entire projector itself. It might even be worthwhile to completely redesign the projector's architecture by increasing its depth and including modern concepts like attention and transformers. Especially vision transformers are known to benefit disproportionately from large amounts of training data, which is usually readily available in the context of contrastive learning.

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
