# OpenReview forum: "Improving Nonlinear Projection Heads using Pretrained Autoencoder Embeddings"
_ICLR.cc/2025/Conference — Submitted to ICLR 2025_

### Official Review · Reviewer_1A9E · 2024-10-21

**Soundness:** 2
**Presentation:** 2
**Contribution:** 2
**Rating:** 3
**Confidence:** 5

**Summary:**

The paper studies in depth about the projection head of SimCLR. The author suggests replacing the conventional projector with a pretrained shallow autoencoder could improve the performance of the trained model.

**Strengths:**

- The paper is well-written and easy to follow.

- The method is quite simple and easy to implement.

- The improvements are quite good

**Weaknesses:**

- First of all, the paper mostly looks like a technical report paper, it lacks the strong idea and results to make it novel.

- The experiments were only conducted with small-scale datasets and lacked a comparison with a large family of self-supervised learning.

- There are no consistent patterns of the number of dimensions, activation function, normalization, etc that could be followed, depending on the dataset, we need to run a bunch of trials to see which combination works best.

- Does the number of layers in the autoencoder affect the performance?

**Questions:**

see Weaknesses

---

> ### Author Response · Authors · 2024-11-25
> **Answer: Style and novelty, dataset and model selection, lack of patterns, autoencoder depth**
>
> 1. **Style and novelty**: It is understandable that the paper might appear in some parts as being more of a technical report than a research paper. However, we also clearly present original research combined with experiments that lead to an analysis of quantifiable results based on our described methods. As already argued in the previous reviews, the novelty of our method lies in offering an easy way to potentially improve, as supported by our empirical evidence, the quality of contrastively learned representations within the SimCLR framework by pre-training autoencoder embeddings without the use of any labeled data. In the context of contrastive learning, we find the latter to be a highly desirable trait of our proposed method, as labeled samples are often scarce and usually expensive to produce.
> 2. **Dataset and model selection**: See 2./3. of previous review.
> 3. **Lack of patterns**: One of the most important patterns can be seen in Table 2, where the AE projector consistently outperforms the default MLP projector. Table 4 illustrates the consistent pattern that narrower AE projectors outperform wider MLP projectors. From Table 5 one can extract the pattern that with the AE projector non-RELU activations outperform RELU activations in most cases. In Table 6 one can observe the pattern, that freezing the weights within the autoencoder embedding consistently improves downstream classification accuracy. All these oberservations are of course empirical (limited to our experiments) and may fluctuate within your specific use case and dataset. You are therefore correct, that you would need to run your own experiments to determine some combination of these factors that would lead to a "globally optimal" (or close to) model. Our observations can however help you to potentially eliminate some less promising MLP architectures like "not using a pre-trained embedding at all", "using a pre-trained embedding with ReLU", "using wider MLP projectors (and wasting compute) while narrower AE projectors might perform better" etc.
> 4. **Autoencoder depth**: Probably yes, as one can expect the embeddings of deeper autoencoders to be of higher quality than of the shallow one used in our paper. This remains an interesting question for future research.

---

> > ### Comment · Reviewer_1A9E · 2024-11-26
> >
> > I still think that the paper is still not meeting the criteria for this venue, so I'll maintain my score.

---

### Official Review · Reviewer_dkLY · 2024-11-01

**Soundness:** 2
**Presentation:** 2
**Contribution:** 2
**Rating:** 5
**Confidence:** 3

**Summary:**

This paper is an empirical study to improve the effectiveness of the standard 2-layer MLP projection head featured in the SimCLR framework by pretrained autoencoder embeddings. The paper’s result show accuracy improvements and dimensionality reductions using the modified nonlinear projection heads in image classification tasks.

**Strengths:**

- An interesting setting in contrastive learning: optimizing projection head performance.
- Investigated the hypothesis of whether pretrained autoencoder embeddings are able to improve the performance of the standard 2-layer MLP projection head used in the SimCLR framework.
- The paper provided empirical evaluation results on several datasets.

**Weaknesses:**

- The background introduction, including SimCLR, autoencoders, pretraining autoencoder embeddings, etc, seems too long in the paper. The author can provide a short summary of this and move the detailed background introduction to the appendix.
- Following the above, the paper lacks a theoretical statement for replacing the SimCLR projection head with autoencoder embeddings. It seems unclear why this approach should theoretically enhance SimCLR's representation quality. The authors reported empirical results in the paper, which is good, but beyond that, it would be good to provide a theoretical analysis.
- The paper included the results on five image datasets. However, it seems unclear why those datasets were selected. The dataset selection could introduce biases, which may impact generalizability.
- It would be good to discuss why SimCLR was selected, as there are more new projection head designs. Or discussions about why the nonlinear nature of the SimCLR projector is beneficial. If SimCLR is not a specific choice, then maybe some ablation study on other projection heads or CL methods can be investigated to see if the findings on SimCLR can be generalized to other models.

**Questions:**

Please see the comments above.

---

> ### Author Response · Authors · 2024-11-25
> **Answers: Length of introduction, lack of theoretical background, dataset and framework selection**
>
> 1. **Length of introduction**: You're certainly correct, as this was our first ICRL submission, we were unsure about how much detail to include in the introduction, related work and the theoretical background regarding autoencoders and the SimCLR framework. We do however hope, that this does not diminish or reduce the value of our core findings presented in this paper.
> 2. **Lack of theoretical background**: Like the previous reviewer, you're also correct in observing the lack of a strong theoretical understanding of why these effects occur. We do however offer some high-level insights and conjectures on some of these effects which might be useful for developing more theoretically minded explanations in the future.
> 3. **Dataset selection**: Dataset _and backbone_ selection was mainly influenced by time and compute constraints. As you probably have noticed, we use ResNet34 over ResNet50 as ResNet34 generates 4x fewer output features (512) compared to ResNet50 (2048). This then also reduces the width of the autoencoder embedding which is used as the projector's input layer. This however is only the embedding, meaning the encoder's input and the decoder's output layer have to remain larger to achieve a contraction and expansion within the network. Using ResNet50, this would have dramatically increased the width and therefore the size of the entire SimCLR architecture exceeding our GPUs VRAM size. In order to compensate for the loss in expressiveness (i. e. model capacity) introduced by choosing the shallower and narrower ResNet34, we decided to also reduce dataset complexity by choosing classification datasets with 10 to 30 classes, with the exception of CIFAR100. Furthermore, contrastive learning with the NT-Xent loss benefits a lot from large batch sizes of up to 4096 (and even larger). In order to allow for larger batch sizes in a resource constrained environment, we chose datasets with a rather small individual sample resolution ranging from 32x32px to 128x128px. This means that our maximum sample resolution was roughly half that of ImageNet, allowing our GPUs to work at full capacity without being overwhelmed by data. In the end, we trained our modified SimCLR architecture on five well-known image classification datasets that met these criteria.
> 4. **Framework selection**: We chose the SimCLR framework primarily as a vehicle to demonstrate our idea using pre-trained autoencoder embeddings in projection heads for contrastive learning applications. Furthermore, the SimCLR framework is not only easy to implement but also recognized as a fundamental concept in the field of contrastive learning. As a reference architecture, SimCLR offers a sound theoretical framework for demonstrating new idea's and conducting experiments in the realm of contrastive learning. However, further investigating the generalizability of our method to other contrastive learning architectures is an exciting question for future research.

---

### Official Review · Reviewer_goFw · 2024-11-03

**Soundness:** 2
**Presentation:** 1
**Contribution:** 2
**Rating:** 3
**Confidence:** 4

**Summary:**

The paper explores the impact of using pre-trained autoencoder embeddings within the SimCLR framework’s projection head to improve the quality of learned representations in self-supervised contrastive learning. The authors propose replacing the standard 2-layer MLP projector’s input layer with a trained autoencoder embedding while applying different activation functions and architectural modifications. They evaluate these changes on five well-known image classification datasets.

**Strengths:**

1. The experiments conducted could be a good reference for industry applications that don't want to fine-tune the whole framework.
2. The method is very simple and direct.

**Weaknesses:**

1. Novelty: The paper lacks a deeper theoretical explanation of why pre-trained autoencoder embeddings enhance performance. The observed benefits are primarily justified through empirical evidence.  Also, similar ideas have been applied to many industry scenarios during the last five years. As long as it is a projection layer, is there too much difference between the MLP layer and AE?

2. The experiments are too limited to make such a big claim. The datasets implemented are mostly STL10, CIFAR10, etc, which is too simple and the scale is too limited to support its claim. Some differences will not stand if scaled up to a larger dataset.

**Questions:**

1. How would your approach scale to larger image datasets or different domains?
2. What are the trade-offs between freezing and fine-tuning the autoencoder embeddings, and could there be scenarios where fine-tuning might be beneficial?

---

> ### Author Response · Authors · 2024-11-25
> **Answer: Scalability and trade-offs between freezing and fine-tuning the autoencoder embeddings**
>
> 1. **Scalability**: For the most part, our approach offers similar scalability characteristics as the regular SimCLR method proposed by Chen et al. The only added overhead is encountered during the pre-training phase of the shallow autoencoder in order to then extract its learned embeddings which are compressed into a single linear layer. As with SimCLR's default MLP projector, increasing the width (or depth) of the autoencoder does indeed increase training and inference time. In both cases, the width of the projector is dictated by the number of output features present in the select backbone. This means that choosing a narrower backbone will also allow you to choose a narrower projection head, decreasing training and inference time at the cost of lower quality representations and a net loss in downstream accuracy. It is important to point out that once the pre-trained linear embedding layer has been extracted from the autoencoder, the user _does not_ experience any additional overhead in training the SimCLR architecture, as by this point, our projector's basic architecture has become identical to SimCLR's default 2-layer MLP projector. The only difference, ignoring our experiments with non-ReLU activation functions, lies solely in the pre-trained weights contained within the projector's input layer.
> 2. **Trade-offs between freezing and fine-tuning embeddings:** The main disadvantage of not freezing the autoencoder embeddings within the projector's input layer is an empirically observed decrease in downstream classification accuracy. As it is not fully clear which dataset or procedure you would like to use for fine-tuning the (pre-trained?) embeddings, we might assume a classical transfer learning approach in which you propose to pre-train autoencoder embeddings on large multipurpose datasets like ImageNet and then fine-tune using a smaller task-specific dataset, i. e. the datasets we used for pre-training our autoencoder embeddings. This approach, especially when combined with a shallow autoencoder and a large dataset like ImageNet, will most likely suffer under the autoencoder's limited expressiveness, i. e. lack of learnable network parameters. However, this fine-tuning approach might be feasible and beneficial when training deeper and more sophisticated projectors, which is an interesting idea for future research but also a rather significant departure from SimCLR's default 2-layer MLP.

---

### Official Review · Reviewer_cMND · 2024-11-03

**Soundness:** 2
**Presentation:** 3
**Contribution:** 2
**Rating:** 3
**Confidence:** 5

**Summary:**

This paper improves the projection head in contrastive learning by incorporating pre-trained autoencoder embeddings. The experimental results validate that the pre-trained autoencoder is beneficial for extracting meaningful representations in the embedding layer.

**Strengths:**

The paper improves the projector by taking advantage of the autoencoder's ability to capture high-quality representations, which improves the performance of the contrastive learning method.

**Weaknesses:**

1) The biggest problem with this article is the lack of innovation. This article just experimentally verifies that swapping the initialization of the projected head in SimCLR for a pre-trained AE is effective. From this point of view, this article is more like an experimental report than an academic paper. I would suggest that the authors could give more insight or theoretical analysis to prove why this works.
2) The currently listed references only include 9 papers, which is an inadequate number. It is recommended that an in-depth analysis of existing related research be conducted further.
3) Unnecessary symbols in abstract and introduction should be avoided, such as projection head $g(\cdot)$ and feature extractor $f(\cdot)$. It is reasonable to introduce them in the method section to formally explain the concepts.
4) The logical narrative in the introduction section needs adjustment. It is better to first point out the role and limitations of existing projection heads. Subsequently, it is reasonable to propose using pre-trained autoencoder embedding as the initial weights for the projection head.
5) The contributions in the introduction should be concise. For instance, the proposal of code should not be listed here.
6) In the final paragraph of related work, it is recommended to briefly discuss the differences between your work and the previously mentioned studies.
7) The experimental design provided is insufficient to explain how the projector is capable of generating high-quality representations. When evaluating the overall experiments, relying solely on classification accuracy on classification tasks as the evaluation metric may not provide a comprehensive understanding of the model's performance. To support the conclusions more robustly, it is advisable to incorporate additional evaluation methods such as t-SNE visualization.

**Questions:**

What is meant by the "high-level structure of the training data" in the discussion section, please give a more detailed description of it.

---

> ### Author Response · Authors · 2024-11-25
> **Answer: Explanation of "high-level structure of the training data"**
>
> For a given (classification) task that involves learning meaningful representations from a fixed (training) dataset, our paper proposes a two step approach for improving the quality of the contrastively learned representations:
>
> 1. We train a shallow autoencoder to capture the "high-level structure of the training data", i. e. basic or surface level features present  in the training dataset. As these features are captured within the embedding layer of a _very shallow_ autoencoder, we assume that those features have to be rudimentary ("high-level") in nature. Assuming otherwise would contradict the basic value proposition of Deep Learning, i. e. deep(er) neural networks being superior in learning generalizable representations from complex datasets.
> 2. We then exchange the randomly initialized weights in the input layer of SimCLR's default projection head with our pre-trained weights representing high-level features that are compressed within the autoencoder's embedding layer. After that we contrastively train our modified SimCLR architecture on the same training dataset as the autoencoder and observe consistent improvements in image classification accuracy on five well-known datasets.
>
> We purposefully choose a shallow autoencoder to demonstrate the positive effect on classification as training this architecture is fast and does not require any labels. Vanilla (and shallow) autoencoders are also easy to implement and offer a sound theoretical framework that can be easily understood. Naturally this leads to the assumption, that training deeper and more sophisticated autoencoder architectures might offer an even greater improvement in downstream classification tasks.

---

> ### Comment · Reviewer_cMND · 2024-11-26
>
> Thank you for your response. After reviewing all the feedback, I’ve decided to keep my score as it is.

---

### Meta-Review · Area_Chair_nEYJ · 2024-12-18

**Metareview:**

This paper presents a method aimed at enhancing contrastive learning by leveraging pretrained autoencoder embeddings as a substitute for nonlinear projection heads, with the promise of empirical advancements.

In terms of strengths, Reviewer cMND notes the method's straightforwardness and ease of implementation, while Reviewer goFw praises the clear and well-organized structure of the paper.

However, the paper's weaknesses are significant. Reviewer 1A9E points out the lack of both innovation and a solid theoretical backing to underpin the empirical findings. Reviewer cMND expresses concerns over a limited dataset selection, which may introduce bias and limit the generalizability of the results. Furthermore, Reviewer goFw and Reviewer dkLY lament the absence of a comprehensive theoretical rationale for replacing the SimCLR projection head with autoencoder embeddings, and they consider the scope of the experiments too narrow to convincingly support the paper's claims.

The decision to reject the paper is primarily driven by the unanimous concerns regarding its theoretical depth, innovation, and the generalizability of its results.

**Additional Comments On Reviewer Discussion:**

During the discussion, Reviewer cMND and Reviewer 1A9E highlighted the paper's lack of theoretical analysis, while Reviewer goFw and Reviewer dkLY voiced reservations about the generalizability of the findings due to a limited dataset selection.

The authors' rebuttal did not successfully address the reviewers' issues. The lack of novelty remained a critical, unresolved concern for Reviewers cMND and 1A9E. As for the choice of datasets, the justification provided by the authors did not alleviate the worries of Reviewers goFw and dkLY about potential bias and generalizability.

In reaching the final decision, the reviewers' concerns about theoretical depth and innovation were weighed heavily. The rebuttal did not sufficiently address these points, thereby solidifying the lean towards rejection.

---

### Decision · Program_Chairs · 2025-01-22

Reject